# Effects of Manual Therapy on Parkinson’s Gait: A Systematic Review

**DOI:** 10.3390/s24020354

**Published:** 2024-01-07

**Authors:** Arnaud Delafontaine, Thomas Vialleron, Gaëtan Barbier, Arnaud Lardon, Mélodie Barrière, María García-Escudero, Laurent Fabeck, Martin Descarreaux

**Affiliations:** 1Department of Orthopedic Surgery, Université Libre de Bruxelles, 1050 Bruxelles, Belgium; laurent.fabeck@ulb.be; 2Department of Sciences of Physical Activity, Université Québec Trois-Rivières, Trois-Rivières, QC G9A 5H7, Canada; melodie.barriere@uqtr.ca (M.B.); martin.descarreaux@uqtr.ca (M.D.); 3Laboratoire Interdisciplinaire en Neurosciences, Physiologie et Psychologie: Activité Physique, Santé et Apprentissages (LINP2), UFR STAPS, Université Paris Nanterre, 92000 Nanterre, France; thomas.vialleron@parisnanterre.fr; 4Institut Franco-Européen de Chiropraxie, 94200 Ivry-sur-Seine, France; gbarbier@ifec.net (G.B.); alardon@ifec.net (A.L.); 5Complexité, Innovation, Activités Motrices et Sportives (CIAMS) Laboratory, Université Paris-Saclay, CEDEX 91405 Orsay, France; 6Complexité, Innovation, Activités Motrices et Sportives (CIAMS) Laboratory, Université d’Orléans, 45067 Orléans, France; 7Faculté de Médecine et des Sciences de la Santé, Université Catholique de Valence, San Vicente Martir, 46900 Valence, Spain; maria.escudero@ucv.es

**Keywords:** manual therapy, Parkinson’s disease, gait, dynamic gait index, gait performances

## Abstract

Manual therapy (MT) is commonly used in rehabilitation to deal with motor impairments in Parkinson’s disease (PD). However, is MT an efficient method to improve gait in PD? To answer the question, a systematic review of clinical controlled trials was conducted. Estimates of effect sizes (reported as standard mean difference (SMD)) with their respective 95% confidence interval (95% CI) were reported for each outcome when sufficient data were available. If data were lacking, *p* values were reported. The PEDro scale was used for the quality assessment. Three studies were included in the review. MT improved Dynamic Gait Index (SMD = 1.47; 95% CI: 0.62, 2.32; PEDro score: 5/10, moderate level of evidence). MT also improved gait performances in terms of stride length, velocity of arm movements, linear velocities of the shoulder and the hip (*p* < 0.05; PEDro score: 2/10, limited level of evidence). There was no significant difference between groups after MT for any joint’s range of motion during gait (*p* > 0.05; PEDro score: 6/10, moderate level of evidence). There is no strong level of evidence supporting the beneficial effect of MT to improve gait in PD. Further randomized controlled trials are needed to understand the impact of MT on gait in PD.

## 1. Introduction

From a medical point of view, gait refers to the human steady state of walking [1]. Simple in appearance, this daily locomotor task is known to be complex for the balance control system. It requires the integration of multiple sensory information from somatosensory, vestibular, and visual systems [2,3,4,5], necessitates the coordination of multiple skeletal muscles and involves executive functions [6,7]. Consequently, populations with sensory, motor, or cognitive deficits may be exposed to gait disorders. Gait performance depends on many biomechanical features that can be observed during gait analysis [8]. Spatiotemporal features (e.g., velocity, step length, stride length, step with, and walking cadence) can be assessed subjectively with functional evaluations by clinicians [9,10], whereas kinetics (e.g., center of mass shift, forces, joint moments) and kinematics parameters (e.g., range of motion (ROM)) can be objectively assessed with biomechanical analysis in a laboratory [11]. Gait is indeed classically used in the literature to investigate the effects of rehabilitation programs on motor performances in neurodegenerative populations, such as stroke [12] or Parkinson’s disease (PD) [13,14,15,16].

PD is a common progressive neurodegenerative disease [17]. It is characterized by different motor and non-motor symptoms that impact the quality of life of patients to a variable degree. The principal clinical signs of the disease include rest tremor, bradykinesia, rigidity and postural instability [18]. Secondary motor symptoms are commonly found in the literature, such as flexed posture and freezing of gait [19,20]. The degradation of motor function leads to difficulties in coping with daily locomotor tasks such as gait initiation [21], walking [7], obstacle crossing [22], or moving around a more confined space of a home [23]. The increased fall risk due to these motor disorders is one of the most reported problems by patients with PD [24,25]. Additionally, PD is generally associated with a large spectrum of non-motor symptoms that also affect gait and functional capacities, such as apathy, depression, cognitive dysfunction, behavioral disorders, sensory abnormalities and sleep disturbances [26,27,28].

Because of the increasing longevity and industrialization, PD is increasing in every major region of the world [29]. The number of patients who will need care in rehabilitation will continue to grow in the coming decades. More precisely, the number of people with PD is projected to double to over 12 million by 2040 [30]. Clinicians should be aware of this trend and be informed of the most efficient practices.

Manual therapy (MT) is a passive movement applied by clinicians (e.g., physiotherapists, osteopaths, chiropractors) on musculoskeletal structures of the patients, such as the joints, soft tissues and nerve tissues [31]. In rehabilitation, MT is principally utilized to treat a wide range of pain disorders, such as neck pain [32,33], low back pain [34], knee osteoarthrosis [35,36], cervical and lumbar radiculopathy [37], plantar heel pain [38] and tension-type headache [39]. MT is also used to stimulate several biomechanical, neurophysiological and psychological changes that could result in motor control improvements [36].

Regarding the role of MT in improving postural stability, balance and motor performance, a limited amount of research has been published [40,41]. In PD, a recent narrative review by Li et al. [42] suggested that MT may be beneficial in addressing motor-related and neurologic symptoms [42].

However, the authors concluded that research in this field remains limited and that more investigations are needed. To date, no systematic review has collected results on the relationship between MT and gait performance in rehabilitation programs.

Hence, the purpose of this article is to analyze the effect of MT on gait in PD by means of a systematic literature review and statistical analyses, comparing the gait outcomes of the intervention groups with the control groups. It will contribute to providing evidence-based practices from scientific data in order to integrate MT in locomotor rehabilitation programs in a reasoned manner.

## 2. Methods

### 2.1. Design and Literature Screening

The Preferred Reporting Items for Systematic Reviews and Meta-Analyses (PRISMA) methodology was employed in this systematic review [43,44]. The relevant databases PubMed, Science Direct, Springer and Sage were used for a systematic literature search for articles published prior to 5 December 2023 with no time limit. In addition, a manual search was conducted using the reference list of selected studies. The keywords used for the search in PubMed were: “manual therapy” AND (gait OR walk). The selection procedure was conducted by two experts in MT (AD and GB). Disagreements were discussed with a third expert (TV) until a mutual consensus was reached. First, a review was performed on all available titles obtained from the literature search with the selected keywords. All relevant or potentially relevant titles were included in the subsequent phase. Then, the abstracts were reviewed with all relevant or potentially relevant articles included in the following phase. Finally, full-text articles were reviewed to ensure that only relevant studies were included. Reference lists of all included articles were reviewed in the same way to possibly include studies through cross-referencing.

### 2.2. Inclusion and Exclusion Criteria

To reach the best level of evidence, we included only randomized controlled trials and controlled clinical trials published in peer-reviewed journals that aimed to explore the effects of MT on gait parameters. We selected only idiopathic PD patients and all MT techniques. Gait could be evaluated by means of functional evaluations and kinematic or kinetic analyses. We included only articles published in English or French. The following exclusion criteria were used: lack of gait assessment, no application of MT, no control group, case report and review. They were also excluded if MT was combined with any intervention not provided to the control group (i.e., not only MT effects were measured).

### 2.3. Data Extraction and Main Measurements Examined

Data were extracted from the selected articles by two authors (AD and GB). The extracted data were checked by two other authors (AL and MD), and disagreements were resolved with a third (TV). The following data were extracted for each selected article: (1) the authors ‘names and the date of publication; (2) the number of subjects involved in the experiment; (3) the PD group details (in the following order: number of participants, mean age, Hoehn and Yahr score, Unified Parkinson’s Disease Rating Scale (UPDRS) total score, MT technique, duration of each session and their frequency); (4) control group details; and (5) the main outcomes related to gait with the main results. When information could not be provided, it was indicated by a “?”.

### 2.4. Quality and Risk of Bias Assessment

The PEDro scale, a valid measure of the methodological quality of clinical trials, was used to assess the risk of bias of the selected studies [45]. The scale was chosen for its ability to provide an overview of the external (criterion 1), internal (criteria 2–9) and statistical (criteria 9 and 10) validity of controlled trials. The scale is divided into 11 criteria, but the first criterion is not calculated in the total score. The output of each criterion could be either “yes” (y), “no” (n) or “do not know” (?). A “y” was given a score of one point, while a “n” or “?” was assigned zero points. Studies with a total score of 5–10/10 (≥50%) were considered to be of high quality, and scores of 0–4/10 (<50%) as low quality. Two evaluators (AD and GB) assessed the quality of the included studies independently. The measures were checked by two other authors (MD and MB). In the event of disagreements, a group discussion was held with a third expert (TV) to reach a mutual consensus.

### 2.5. Statistical Analyses

Estimates of effect sizes (comparing the intervention groups and the control groups) accompanied with a measure of statistical uncertainty (95% confidence interval [95% CI]) were calculated for each outcome when sufficient data were reported. Estimates of effect sizes were reported by standard mean difference (SMD) and their respective 95% CI. The magnitude of the overall effect was quantified as trivial (<0.2), small (0.2–0.49), moderate (0.5–0.79) or large (≥0.8) [46], [47]. When data were lacking to calculate estimates of effect sizes, exact *p* values were reported.

### 2.6. Level of Evidence

The strength of evidence of primary outcomes was established as described by Van Tulder et al. (2003) [48]. The method is based on effect size estimates with a measure of statistical uncertainty (SMD; 95% CI), statistical heterogeneity (I^2^) when applicable (multiple studies) and risk of bias (PEDro scale). The level of evidence was considered strong, with consistent findings among multiple high-quality randomized controlled trials (at least two trials with a PEDro score ≥5/10 that were statistically homogenous: I^2^ *p* ≥ 0.05). The level of evidence was considered moderate, with consistent findings among multiple low-quality randomized controlled trials and/or clinical controlled trials (two trials with a PEDro score <5/10 that were statistically homogenous) and/or one high-quality randomized controlled trial. The level of evidence was considered limited when only one low-quality trial was identified. The level of evidence was conflicting when there was inconsistency among multiple trials (I^2^ *p* < 0.05).

## 3. Results

### 3.1. Included Studies

A total of 425 titles were screened in the first search stage; one more was included through cross-referencing, and 321 were excluded because they did not concern our research question. Following exclusions, 105 studies were considered for an abstract review. A further 100 were excluded in this second stage because they did not meet the inclusion criteria. Finally, five full-text articles were assessed for eligibility, and three of them were included in this systematic review [49,50,51]. The flow diagram of the current systematic review is provided in Figure 1, and a summary of the selected studies is provided in Table 1. The results from the quality assessments are provided in Table 2. According to the PEDro Scale, two studies obtained a high-quality methodology score [49,51] and one study was rated as low-quality [50]. The mean score was 4.33 ± 2.08 (ranging from 2 to 6).

### 3.2. Characteristics of the Population

The three studies included a total of 89 patients with idiopathic PD. The average sample size was 29.67 ± 10.60 (ranging from 20 [50] to 41 subjects [51]). Additionally, two studies included a total of 51 healthy subjects in their control groups, with an average of 25.5 ± 24.75 (ranging from 8 [50] to 43 subjects [51]). One study did not mention the mean age of the participants but the range (45 to 68 years) [50]. The mean age of the patients in the two other studies was 66.94 ± 8.43 years [49,51]. Healthy participants were age-matched, with a mean age of 66.77 ± 9.50 [51].

The Hoehn and Yahr scores averaged 1.92 ± 0.05 in one study [51]. Another study used the modified Hoehn and Yahr scale with an average score of 2.44 ± 0.01 [49]. In both studies, the UPDRS total score averaged 27.71 ± 10.44. In the last study, only the UPDRS motor score was reported with an average of 14.30 [50]. Altogether, these results suggest that patients from the included studies can be classified as moderately affected and physically independent.

### 3.3. Characteristics of the Interventions

One study used lumbosacral mobilization [49], and two studies evaluated osteopathic manipulative treatment (OMT) [50,51]. Lumbosacral mobilization was performed manually, applying Cyriax mobilization techniques to the intervention group for one session of 10 min. In the study of Wells et al. [50], OMT consisted of techniques that aimed to improve flexibility, muscle length and mobility of the spine. The single session lasted approximately 30 min. The study of Terrell et al. [51] included two standardized OMT protocols, OMT neck down (OMT-ND) and OMT whole-body (OMT-WB). The first targeted the cervical, thoracic, and lumbar spine, shoulder girdle, sacroiliac joint, hip bones, leg musculature, and ankles. The second included all the techniques in the OMT-ND protocol but also techniques that targeted cranial dysfunction, and both lasted approximately 25 min. Two studies included sham interventions with patients [50] and healthy participants [50,51]. Sham protocols consisted of examinations of the subject’s active and passive ROM in the spine and extremities, testing the same joints that were treated with OMT and lasting similar durations to other treatments.

### 3.4. Effects of Interventions in Different Outcomes

Dynamic Gait Index: the scale was used by Seçkinoğulları et al. [49]. Intergroup comparisons showed a significant difference in favor of the intervention group with a large effect size (SMD = 1.47; 95% CI: 0.62, 2.32). The study was identified as a randomized controlled trial and obtained a PEDro score of 5/10; thus, the level of evidence for this outcome was moderate.

Joint ROM during gait: two studies evaluated the effect of OMT on joint ROM during gait [50,51]. However, because of the lack of data in one study [50], no meta-analysis could be performed. Both studies found no significant difference between groups after treatments. Wells et al. [50] only reported *p* values > 0.05 for the hip, knee and ankle ROM during gait (data were lacking to calculate effect sizes). In the Terrell et al. [51] study, patients with PD and healthy controls showed significant differences in pretreatment ROM, with large effect sizes, for hip (SMD = 0.97; 95% CI: 0.51, 1.42) and knee (SMD = 0.88; 95% CI: 0.43, 1.33). However, there was no significant difference between PD and healthy participants in pretreatment ROM for ankles (SMD = 0.20; 95% CI: −0.23, 0.63). Regarding patients with PD in posttreatment conditions, there were no significant differences between OMT-WB and the sham group for hip ROM (SMD = −0.24; 95% CI: −1.00, 0.53), as well as between OMT-ND and the sham group for hip ROM (SMD = −0.18; 95% CI: −0.95, 0.60). There was also no significant difference between OMT-ND and the sham group for knee ROM (SMD = −0.58; 95% CI: −1.37, 0.21). OMT-WB and the sham group were significantly different at baseline for knee ROM (SMD = −0.88; 95% CI: −1.68, −0.08), so both groups were not compared in posttreatment conditions for this outcome. Finally, no significant differences were found between OMT-WB and the sham group for ankle ROM (SMD = −0.46; 95% CI: −1.23, 0.31), as well as between OMT-ND and the sham group for ankle ROM (SMD = −0.12; 95% CI: −0.89, 0.66). The study was identified as a randomized controlled trial and obtained a PEDro score of 6/10; thus, the level of evidence for these outcomes was moderate.

Spatiotemporal variables: for these outcomes, no sufficient data were reported to calculate estimates of effect sizes, so only *p* values are communicated. Wells et al. [50] showed that stride length was significantly improved in patients with PD treated with OMT when compared with healthy subjects treated with OMT (*p* < 0.048) and patients treated with the sham protocol (*p* < 0.022). The authors also reported greater maximum linear velocities of the shoulder and the hip in patients with PD treated with OMT than in healthy subjects who received OMT (*p* < 0.028 and *p* < 0.03, respectively) and sham-treated patients (*p* < 0.0009 and *p* < 0.007, respectively). Finally, the velocity of arm movements was significantly improved in patients with PD after OMT when compared with patients in the sham group (*p* = 0.001). Because of the lack of data and the low quality of the study (PEDro score of 2/10), the level of evidence was limited for these outcomes.

## 4. Discussion

The aim of this systematic review was to determine the effects of MT on gait in patients with PD. Three studies and two techniques of MT (i.e., lumbosacral mobilization and OMT) were identified. The results showed no strong level of evidence supporting the beneficial effect of MT to improve any gait outcome. The major obstacles in establishing strong levels of evidence (and in conducting meta-analyses) were the lack of data and the heterogeneity in gait variables. The results obtained for each outcome are discussed in detail below.

### 4.1. Dynamic Gait Index

Only one study used a scale to evaluate gait [49]. The Dynamic Gait Index is a valid, reliable and recommended tool in the evaluation of gait in PD [52,53]. The scale consists of eight different gait tasks, such as walking on a level surface, changing gait speed, turning the head in a horizontal and vertical direction while walking, rapid directional changes, stepping over and around an obstacle, and stair climbing. A score of 19 or less is associated with an increased risk of falling (the maximum score is 24 [52]). A modified version of the scale, with an expanded scoring system, has been recently developed to assess dynamic balance more accurately [54]. At baseline, participants of the Seçkinoğulları et al. [49] study had an average score of 19.57 ± 2.76 in the control group and 20.57 ± 2.59 in the intervention group, indicating that these participants, having an index score superior to the cutoff score of 19, were at lower risk of falling. Interestingly, statistical analysis revealed that the score was improved with a large effect size (SMD = 1.47; 95% CI: 0.62, 2.32) in favor of the intervention group. That result suggests that lumbosacral mobilization has the potential to improve the functional capacities of patients in daily locomotor tasks and reduce the fall risk from the early stage to the mid-stage of the disease, when patients are still physically independent. This result is in accordance with a recent study showing a significant relationship between the patients’ ability to move and the risk of falling [55]. PD patients who rated their mobility as good or very good were at a low risk of falls. Maintaining independence and limiting falls are major preoccupations in PD [24,25]. Thus, further randomized controlled trials are needed to study the effect of MT in other clinical tests such as the 6 min walk test, the 10 m walk test, the Timed Up-and-Go test, the Functional Reach test or the Tinetti test, which are also recommended or suggested in the assessment of gait and balance PD [52,55].

### 4.2. ROM during Gait

Adequate ROM is necessary for the successful completion of many daily life activities [56]. For example, in young, healthy adults, experimental restrictions of ROM of the postural chain with orthoses can induce instability and poorer motor performance during locomotor tasks such as gait initiation [57,58,59] and seat-to-stand [60,61]. It is well established that ROM significantly decreases with aging as well as in neurodegenerative diseases [62,63,64,65,66]. Consequently, stretching is regularly recommended in rehabilitation programs to maintain or improve ROM and reduce stiffness in patients with PD [67]. Thus, MT can be considered a complementary approach to maintain or improve ROM. However, a recent systematic review showed that no strong level of evidence supports the beneficial effect of using stretching alone to improve gait outcomes despite some scattered results [68]. These findings are in line with the current research, suggesting that MT alone may not be sufficient to increase ROM during gait as only isolated improvements were identified. For example, Terrell et al. [51] reported a within group significant difference in patients with PD. The administration of the OMT-WB protocol improved hip ROM during gait following treatment compared to pretreatment (*p* = 0.038), but no significant posttreatment joint angle differences were found between groups [51]. Interestingly, the authors also showed that hip and knee ROM during gait were significantly reduced in patients with PD compared to aged-matched healthy subjects, but statistical analysis found no significant difference for the ankle ROM. In contrast, a recent study showed that patients with PD may exhibit a passive ankle ROM that is significantly lower compared to age-matched healthy adults [16]. These findings suggest that passive ROM and ROM during gait are two different variables that may need further investigation in PD.

### 4.3. Spatiotemporal Variables

Only one study explored the effects of OMT on spatiotemporal outcomes, such as stride length, cadence and velocities of the limbs [50]. Wells et al. [50] reported that with OMT treatment, patients ambulated with longer strides and had an improved arm swing, and the magnitude of these changes was significantly greater than in the control groups. Additionally, cadence increased in patients after OMT compared with pretreatment values, but this change was not significantly different from participants in the control groups (*p* < 0.073). Such results suggest that part of the increase in this variable observed in the intervention group could probably be assigned to a placebo effect. Such confounding effects are not surprising since it is known that PD patients are particularly influenced by the belief in improvement [69]. These findings are congruent with a recent randomized study [16], where a single session of stretching was efficient to increase the capacity to generate forward propulsive forces in patients with PD. These encouraging results regarding stride length and velocity, as well as the significant improvement in the Dynamic Gait Index [49], suggest that further randomized controlled trials are needed to explore the impact of stretching or MT on gait performances in PD.

### 4.4. Other Treatments

If MT showed inconsistent results in improving gait parameters, other rehabilitation programs have demonstrated their effectiveness. A recent meta-analysis provided a complete overview of the evidence for the effectiveness of different interventions in the management of PD [70]. The interventions showing benefits on gait parameters included conventional therapy, treadmill training, strategy training (including cueing), dance, martial arts, aerobic exercises, Nordic walking, resistance training, virtual reality, and balance and gait training. Multicomponent programs are particularly efficient in improving motor performance in PD patients [71]. Finally, PD does not affect every patient the same way; thus, it is important to select personalized rehabilitation programs depending on motor and non-motor symptoms, as well as the general health of a patient [55,72].

### 4.5. Limitations of the Study

To establish the level of evidence, a measure of the statistical heterogeneity is normally needed. Because of the lack of data or because no studies used similar variables, we were not able to perform statistical analyses with multiple studies (i.e., meta-analyses). Thus, our confidence in the results must be taken with caution.

To reach the best level of evidence and perform relevant statistical analyses, we choose to include only randomized or clinical controlled trials in the current systematic review. Consequently, some potentially interesting results could have been excluded. For example, A longitudinal study with 21 PD patients found significant improvements in movement restriction using Anma massage [73]. After a single session of 40 min, muscle stiffness, movement difficulties, pain and fatigue were reduced. Moreover, gait speed and stride length were significantly improved after the treatment. These encouraging results suggest that the Anma technique, which shares some similarities with OMT, can be a point of further research to deal with gait impairment in PD. The current literature review was focused on gait parameters. Nevertheless, independence is compromised in PD, and physical autonomy can be assessed by numerous scales and questionaries that do not specifically evaluate gait performance. For example, one cross-over study of 11 PD patients that explored the effects of OMT on motor function was excluded because of the lack of gait variables [74]. It is worth noting that two sessions of OMT per week for 6 weeks significantly improved the MDS-UPDRS score, and thus, motor function.

Globally, there are remaining uncertainties regarding the findings of the included studies [49,50,51], as the nature of original studies raises questions regarding the causal pathways of the various applied treatments and the observed changes. Further studies involving the use of placebo control groups and assessment of patients’ expectations may lead to different interpretations regarding the effectiveness of manual therapy techniques [75]. 

Moreover, Terell et al. [51] showed that the use of osteopathic cranial manipulative medicine may improve gait kinematics in PD patients. However, as highlighted by Guillaud et al. [76] in a systematic review, caution is warranted when interpreting these results due to the absence of reliable diagnostic procedures and the lack of effectiveness. Quality assessment with the Pedro Scale reinforced this caution because items concerning participant or assessor blinding were not fulfilled in this article (see Table 2). 

## 5. Conclusions

Three studies were identified, involving a total of 89 subjects. Despite some improvements, statistical analyses showed no strong level of evidence supporting the beneficial effect of using MT alone to improve gait outcomes in rehabilitation programs. The major obstacle in conducting statistical analyses and establishing strong levels of evidence was the lack of data. Because the effects of MT are not clear, further randomized controlled trials of good quality are needed to understand the impact of MT on gait in PD. Currently, MT is more recommended to treat pain disorders than to improve gait parameters and should be integrated into a complement of multicomponent rehabilitation programs. In fact, multidisciplinary intensive rehabilitation treatments could potentially impede the advancement of motor decline, postpone the requirement for heightened drug intervention, and manifest a neuroprotective impact [77].

## Figures and Tables

**Figure 1 sensors-24-00354-f001:**
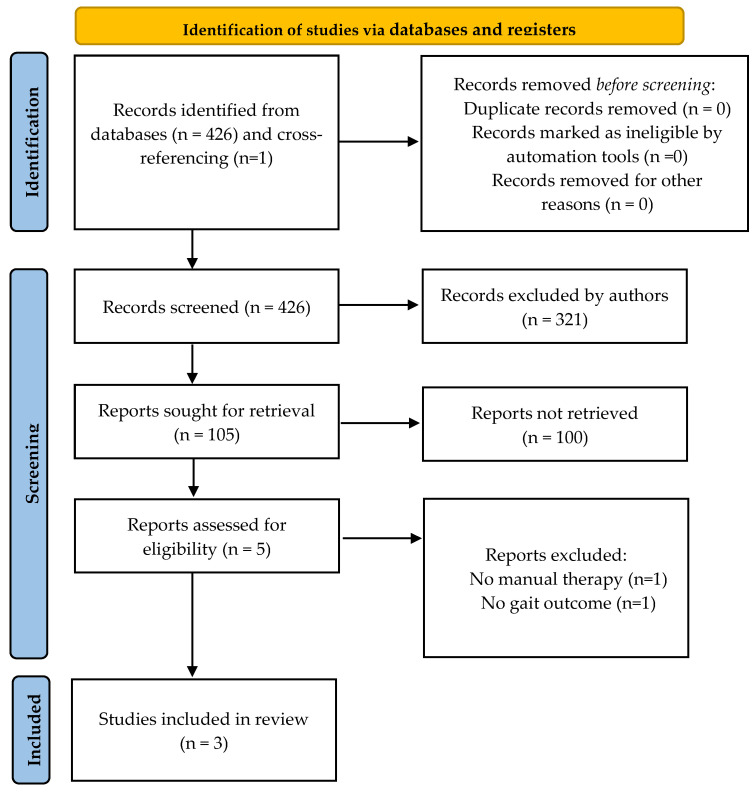
Identification of studies in databases.

**Table 1 sensors-24-00354-t001:** Descriptive table of the included studies.

Studies	Population	Intervention Groups	Control Groups	Outcomes and Main Results
Seçkinoğulları et al. [49]	28 patients with idiopathic PD	*n* = 14 (67.71 ± 5.46 years; mH&Y: 2.23 ± 0.37; UPDRS: 38.14 ± 13.53), lumbosacral mobilization, 10 min, 1 session.	*n* = 14 (65.42 ± 9.01 years; mH&Y: 2.57 ± 0.38; UPDRS: 41.70 ± 12.30), no intervention.	Significant improvement after lumbosacral mobilization compared to control group for Dynamic Gait Index (SMD = 1.47; 95% CI: 0.62, 2.32).
Terrell et al. [51]	41 individuals with idiopathic PD (intervention groups) and 43 age-matched healthy participants (control groups).	*n* = 15 (67.9 ± 12.00 years; H&Y: 1.97 ± 0.70; UPDRS: 19.90 ± 11.00), OMT-WB, 25–30 min, 1 session. *n* = 14 (70.20 ± 8.00 years; H&Y: 1.68 ± 0.80; UPDRS: 14.70 ± 8.40), OMT-ND, 20–25 min, 1 session. *n* = 12 (63.50 ± 7.70 years; H&Y: 2.13 ± 0.70; UPDRS: 24.10 ± 7.00), sham, 20–25 min, 1 session.	*n* = 15 (66.90 ± 11.00 years), OMT-WB, 25–30 min, 1 session. *n* = 15 (68.20 ± 9.50 years), OMT-ND, 20–25 min, 1 session. *n* = 13 (65.20 ± 8.00 years), sham, 20–25 min, 1 session.	PD and healthy controls were significantly different in pretreatment for hip (SMD = 0.97; 95% CI: 0.51, 1.42) and knee ROM (SMD = 0.88; 95% CI: 0.43, 1.33). No significant difference between PD and control in pretreatment for ankle ROM (SMD = 0.20; 95% CI: −0.23, 0.63). In PD, no significant difference between OMT-WB and sham for hip ROM (SMD = −0.24; 95% CI: −1.00, 0.53). No significant difference between OMT-ND and sham for hip ROM (SMD = −0.18; 95% CI: −0.95, 0.60). No significant difference between OMT-ND and sham for knee ROM (SMD = −0.58; 95% CI: −1.37, 0.21). No significant difference between OMT-WB and sham for ankle ROM (SMD = −0.46; 95% CI: −1.23, 0.31). No significant difference between OMT-ND and sham for ankle ROM (SMD = −0.12; 95% CI: −0.89, 0.66).
Wells et al. [50]	20 patients with idiopathic PD and 8 age-matched healthy participants (control group)	*n* = 10 (? years; H&Y: ?; UPDRS: ?), OMT, 30 min, 1 session. *n* = 10 (? years; H&Y: ?; UPDRS: ?), sham, 30 min, 1 session.	*n* = 8 (? years), OMT, 30 min, 1 session.	Significant improvement of stride length in PD after OMT compared with healthy participants and sham (*p* < 0.048 and *p* < 0.022, respectively). Significant improvement of maximum linear velocity of the shoulder in PD after OMT compared with healthy participants and sham (*p* < 0.028 and *p* < 0.0009, respectively). Significant improvement of the velocity of arm movements in PD after OMT compared with sham (*p*= 0.001). Significant improvement of the maximum linear velocity of the hip in PD after OMT compared with healthy participants (*p* < 0.031) and sham (*p* < 0.007).

OMT-WB: osteopathic manipulative treatment whole-body; OMT-ND: OMT neck down; ROM: range of motion; PD: Parkinson disease; H&Y: Hoehn and Yahr Scale; mH&Y: modified Hoehn and Yahr Scale; UPDRS: Unified Parkinson’s Disease Rating Scale; “?”: no available data.

**Table 2 sensors-24-00354-t002:** Quality assessment of the included studies.

Studies	Items by Number on the PEDro Scale											Total Score
	1	2	3	4	5	6	7	8	9	10	11	
**Seçkinoğulları et al.** [49]	y	y	y	y	n	n	n	n	n	y	y	5
**Terrell et al.** [51]	y	y	y	n	n	n	n	y	y	y	y	6
**Wells et al.** [50]	n	n	n	n	y	n	n	n	n	y	n	2

n: criterion not fulfilled; y: criterion fulfilled; total score: each item (except the first) contributes 1 point to the total score, yielding a PEDro scale score that can range from 0 to 10.

## Data Availability

No new data were created.

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
