# Peer review of "Effects of Manual Therapy on Parkinson’s Gait: A Systematic Review"

_sensors, 2024, doi:10.3390/s24020354_

Round 1
Reviewer 1 Report
Comments and Suggestions for Authors
The authors provided a comprehensive overview about the effects of manual therapy on Parkinson’s gait. A systematic review of clinical controlled trials was conducted. This review article has been well written and interesting, but there are several deficiencies. There are only three studies included in the review. It will not be convincing to draw conclusions. In the section "Conclusions", the outlook on manual therapy on Parkinson’s gait are unclear. Therefore, I do not consider this manuscript is worthy of publication in Sensors.
Comments on the Quality of English LanguageMinor editing of English language required.
Author Response
Dear Guess Editor, Editor and Reviewers,
Thank you very much for giving us your feedback and advice on how to improve our article. We have revised the manuscript accordingly. Please find below our point-by-point replies to all of your comments and suggestions. All the modifications that we have made to the original manuscript and iconographies are detailed with our responses. Within the manuscript, the changes are highlighted in red to make them easier to view.
Point to point reply to Reviewer #1
The authors provided a comprehensive overview about the effects of manual therapy on Parkinson’s gait. A systematic review of clinical controlled trials was conducted. This review article has been well written and interesting, but there are several deficiencies. There are only three studies included in the review. It will not be convincing to draw conclusions. In the section "Conclusions", the outlook on manual therapy on Parkinson’s gait are unclear. Therefore, I do not consider this manuscript is worthy of publication in Sensors.
Reply: we greatly thank the reviewer for scrutinizing our manuscript and for his/her positive appreciation of our work. Regarding the numbers of studies included in the review, please note that this is not a publication criterion regarding the current journal recommendations and PRISMA guidelines. The objective of the current study is to provide a state-of-art and research perspectives regarding the rehabilitation of patients with Parkinson's disease. Our discussion and conclusion highlight that there are encouraging results supporting the beneficial effects of using manual therapy to improve gait outcomes in patients with Parkinson’s disease and that further randomized controlled trials of good quality are needed to recommend its use with a higher level of evidence. This kind of conclusion is usual in a systematic review when studying a field in which a limited amount of research has been published. For example, see our work on the impact of stretching on gait parameters (Vialleron et al., 2020, reference number [67]).
Please also see, a new paragraph about the effects of multidisciplinary intensive rehabilitation treatments in Parkinson Disease and a new reference, line 182-185: In fact, multidisciplinary intensive rehabilitation treatments could potentially impede the advancement of motor decline, postpone the requirement for heightened drug intervention, and manifesting a neuroprotective impact [75].
Point to point reply to Reviewer #2
We greatly thank the reviewer for scrutinizing our manuscript and for the relevant comments that helped us further clarify our thoughts. We feel that our manuscript was improved. Please find below the point-to-point reply to each of the remarks raised by the reviewer.
The authors present a timely important review of the effect of manual therapy on Parkinson's gait. The manuscript is nicely composed, but can be improved further in several ways as mentioned below:
1) Pg 1 Line 43, de disease please correct
Reply: we greatly thank the reviewer for his vigilance. The following sentence was corrected (the new text is highlighted in red):
“The principal clinical signs of the disease include rest tremor, bradykinesia, rigidity and postural instability [14].”
2) It is better to elaborate the acronyms in the introduction, even if it is elaborated in the abstract
Reply: acronyms were elaborated from their first occurrence. To facilitate reading of the manuscript, we chose to limit the number of acronyms.
3) Pg 2 line 72, please explain gait performance.
Reply: the following precision is added in the introduction to clarify this point (the new text is highlighted in red), lines 37-42:
Gait performance depends on many biomechanical features that can be observed during gait analysis [8]. Spatiotemporal features (e.g., velocity, step length, stride length, step with, walking cadence) can be assessed subjectively with functional evaluations by clinicians [9], [10] whereas kinetics (e.g., center of mass shift, forces, joint moments) and kinematics parameters (e.g., range of motion (ROM)) can be objectively assessed with biomechanical analysis in a laboratory [11].
4) It is unconventional to have materials and methods in a review paper. Authors are encouraged to cross-check and remove such sections.
Reply: as asked by the Sensors journal, our systematic reviews follow the PRISMA guidelines (please see reference [40]). To ensure that our review is transparent and complete, methods should be reported in sufficient details, such as search strategy, eligibility criteria, selection process, data collection process, risk of bias assessment and effect measures. To avoid any confusion, we suppressed Materials in the title, and we corrected by (the new text is highlighted in red): 2. Methods.
5) Lines 79 to 93 are useless since it not related to the goals of the paper
Reply: this paragraph explains the search strategy. To present the full search strategy is strongly recommended by PRISMA guidelines and the Sensors Journal.
6) Lines 162 to 168 are useless as well. It's immaterial to state, how many papers the authors have read, and how many selected.
Reply: this paragraph presents the search strategy results. This is also recommended by PRISMA guidelines to enhance the transparency of the systematic review, improve replicability, and enable a review to be more easily updated (please see reference [40]).
7) Fig 1 seems to be not necessary
Reply: Figure 1 is the updated PRISMA Flow Diagram that summarize the search strategy. As stated by PRISMA guidelines, it is highly recommended to include the figure in every systematic review.
8) Table 1, Wells et al., column 3, what's the meaning of the question marks?
Reply: the question mark was used when data were not provided in the included studies. We explained it in the methodology, but we forgot to mention that in the caption. We greatly thank the reviewer for his vigilance. In the legend of Table 1 we added the following precision point (the new text is highlighted in red): “?”: no available data.
9) To my surprise, the manuscript has only 1 figure, and that's highly unconventional. Please try to accommodate at least 5 more different figures. If that's not possible, it is not useful to publish such a review in a journal.
Reply: due to the lack of data and the heterogeneity of the studies, we were not able to perform metanalyses that would have increased the number of figures. It did not seem relevant to us to add more figures to illustrate our results. Please note that the number of figures is note a publication criterion regarding the current journal recommendations and PRISMA guidelines.
10) An extensive re-work is required. The data you have provided seems important though.
Reply: we greatly thank the reviewer for his positive appreciation on our work. We did our best to improve the manuscript by integrating his relevant remarks while maintaining PRISMA guidelines as request by the current journal. We have already published systematic reviews following these guidelines (for example, please see Vialleron et al., 2020, reference number [67]).
Point to point reply to Reviewer #3
We greatly thank the reviewer for scrutinizing our manuscript and for the relevant comments and suggestions that helped us improve the manuscript. Please find below the point-to-point reply to each of the remarks raised by the reviewer.
Comments and Suggestions for Authors
The analysis seems to have some conceptual problems, in the first instance the three studies selected to analyze the effectiveness of the intervention, applied in two variations of the pathology. two groups included with symptoms, The most common symptoms of idiopathic Parkinson’s are tremor, rigidity and slowness of movement. the other study includes participants with other types of Parkinson or parkinsonism.
Reply: we have checked all the selected studies and we can confirm that only patients with idiopathic Parkinson’s disease with similar motor symptoms were included in the current study. The following precisions were added in table 1 (line 3, column 2), in the inclusion and exclusion criteria section, and in the results section in the paragraphs concerning the characteristics of the population (the new text is highlighted in red):
“We selected only idiopathic PD patients and all MT techniques”.
“The 3 studies included a total of 89 patients with idiopathic PD”.
“The Hoehn and Yahr scores averaged 1.92±0.05 in one study [51]. Another study used the modified Hoehn and Yahr scale with an average score of 2.44±0.01 [49]. In both studies, the UPDRS total score averaged 27.71±10.44. In the last study, only the UPDRS motor score was reported with an average of 14.30 [50]. Altogether, these results suggest that patients from the included studies can be classified as moderately affected and physically independent.”
The data of the authors in table 1 and table 2 do not coincide, which are then the authors included in the analysis (?). The data of number of patients of the article of Terrell et al. (2022) do not coincide with the original article (DOI: 10.1515/jom-2021-0203), additionally in this study the typologies of the included pathologies are not clearly identified ().
Reply: we greatly thank the reviewer for his vigilance. In table 2, there was a confusion between the first and the last name of the author. In agreement with a remark of another reviewer, the reference was corrected as follow (the new text is highlighted in red): Terrell et al.
Moreover, data of the article were checked and now coincide with the data in Table1. One standard deviation was corrected (UPDRS: 24.10±7.00). Finally, it is worth to note that a total of 90 individuals were initially recruited for the study, but the final sample sizes included 41 individuals with PD and 43 age-matched controls, as described in table 1.
It is recommended to perform a detailed general review of the analysis and verify if the data included are correctly recorded or if there are data from different articles.
Reply: we confirm that all the data were checked and that they coincide with the original articles. Additionally, we repeated the statistical analyses to confirm our results.
Point to point reply to Reviewer #4
Comments and Suggestions for Authors
The manuscript sheds light on the effects of manual therapy on Parkinson’s gait using a systematic review method. The manuscript is methodologically sound without a reasonable amount of data and largely meets the requirements of the journal.
We greatly thank the reviewer for scrutinizing our manuscript, for his/her positive appreciation on our work and for the relevant comments. We feel that our manuscript was improved. Please find below the point-to-point reply to each of the remarks raised by the reviewer.
Comment 1, Page 7 Line 1 (Table 2).
Authors' names should be listed as “Terrell et al.”
Reply: this correction was applied to the three studies as follow (the new text is highlighted in red in table 2):
SeçkinoÄŸulları et al.
Terrell et al.
Wells et al.
Comment 2, Page 7 Line 1 (Table 2).
Isn't Wells et al. (1999)'s total score 2?
The Pedro scale is divided in 11 criteria, but the first criterion is not calculated in the total score. Reply: we greatly thank the reviewer for his vigilance. After taking into account the next comment, we confirm that the total score for this study is 2. It is worth to note that a “y” gives a score of one point, while a “n” is assigned to zero point, and that the first criterion is not calculated in the total score.
Comment 3, Page 7 Line 1 (Table 2).
Isn't the evaluation result of number 11 of Wells et al. (1999) "No"?
Reply: we greatly thank the reviewer for his vigilance. Initially, this criterion was considered to have been met as point measures and/or measures of variability may be provided graphically. However, the graphics and the data in the study of Wells et al. (1999) did not permit us to calculate a measure of the size of the treatment effect. So, the result of criterion number 11 should be “n”. The correction is highlighted in red in table 2 and in the text where needed.
Point to point reply to Reviewer #5
Comments and Suggestions for Authors
Read the article: Wilczyński J, Ścipniak M, Ścipniak K, Zieliński R, Sobolewski P, Wilczyński I. Assessment of Risk Factors for falls in patients with Parkinson's disease. BioMed Research International. Article ID 5531331 2021; and consider citing it in the Discussion.
Reply: We greatly thank the reviewer for scrutinizing our manuscript and for his/her recommendation. We feel that our manuscript was improved. Please find below our modifications in the discussion section after taking account the reviewer suggestion (the new text is highlighted in red):
Lines 95-102: This result is in accordance with a recent study showing a significant relationship between the patients’ ability to move and the risk of falling [55]. PD patients who rated their mobility as good or very good were at a low risk of falls. Maintaining independence and limiting falls are major preoccupations in PD [24], [25]. Thus, further randomized controlled trials are needed to study the effect of MT in other clinical tests such as the 6-minute walk test, the 10-meter walk test, the Timed Up-and-Go test, the Functional Reach test or the Tinetti test, which are also recommended or suggested in the assessment of gait and balance PD [52], [55].
Lines 147-149: Finally, PD does not affect every patient the same way, thus, it is important to select personalized rehabilitation programs depending on motor and non-motor symptoms, as well as general health of a patient [55], [72].
Please also see, a new paragraph about the effects of multidisciplinary intensive rehabilitation treatments in Parkinson Disease and a new reference, line 182-185: In fact, multidisciplinary intensive rehabilitation treatments could potentially impede the advancement of motor decline, postpone the requirement for heightened drug intervention, and manifesting a neuroprotective impact [75].
Reviewer 2 Report
Comments and Suggestions for Authors
The authors present a timely important review of the effect of manual therapy on Parkinson's gait. The manuscript is nicely composed, but can be improved further in several ways as mentioned below:
1) Pg 1 Line 43, de disease please correct
2) It is better to elaborate the acronyms in the introduction, even if it is elaborated in the abstract
3) Pg 2 line 72, please explain gait performance.
4) It is unconventional to have materials and methods in a review paper. Authors are encouraged to cross-check and remove such sections.
5) Lines 79 to 93 are useless since it not related to the goals of the paper
6) Lines 162 to 168 are useless as well. It's immaterial to state, how many papers the authors have read, and how many selected.
7) Fig 1 seems to be not necessary
8) Table 1, Wells et al., column 3, what's the meaning of the question marks?
9) To my surprise, the manuscript has only 1 figure, and that's highly unconventional. Please try to accommodate at least 5 more different figures, If that's not possible, it is not useful to publish such a review in a journal.
10) An extensive re-work is required. he data you have provided seems important though.
Comments on the Quality of English Language
No comments
Author Response

(The authors gave the same response as above.)

Reviewer 3 Report
Comments and Suggestions for Authors
The analysis seems to have some conceptual problems, in the first instance the three studies selected to analyze the effectiveness of the intervention, applied in two variations of the pathology. two groups included with symptoms, The most common symptoms of idiopathic Parkinson’s are tremor, rigidity and slowness of movement. the other study includes participants with other types of Parkinson or parkinsonism.
The data of the authors in table 1 and table 2 do not coincide, which are then the authors included in the analysis (?). The data of number of patients of the article of Terrell et al.(2022) do not coincide with the original article (DOI: 10.1515/jom-2021-0203), additionally in this study the typologies of the included pathologies are not clearly identified ().
it is recommended to perform a detailed general review of the analysis and verify if the data included are correctly recorded or if there are data from different articles.
Author Response

(The authors gave the same response as above.)

Reviewer 4 Report
Comments and Suggestions for Authors
The manuscript sheds light on the effects of manual therapy on Parkinson’s gait using a systematic review method. The manuscript is methodologically sound without a reasonable amount of data and largely meets the requirements of the journal.
Comment 1, Page 7 Line 1 (Table 2).
Authors' names should be listed as “Terrell et al.”
Comment 2, Page 7 Line 1 (Table 2).
Isn't Wells et al. (1999)'s total score 2?
Comment 3, Page 7 Line 1 (Table 2).
Isn't the evaluation result of number 11 of Wells et al. (1999) "No"?
Author Response

(The authors gave the same response as above.)

Reviewer 5 Report
Comments and Suggestions for Authors
Read the article: Wilczyński J, Ścipniak M, Ścipniak K, Zieliński R, Sobolewski P, Wilczyński I. Assessment of Risk Factors for falls in patients with Parkinson's disease. BioMed Research International. Article ID 5531331 2021; and consider citing it in the Discussion.
Author Response

(The authors gave the same response as above.)

Round 2
Reviewer 1 Report
Comments and Suggestions for Authors
The article analyzes the effect of MT on gait in PD by means of a systematic literature review and statistical analyses, comparing the gait outcomes of the intervention groups with the control groups. This work is relatively comprehensive and logical, and I think it can be published on Sensors.
Author Response
Dear Guess Editor, Editor and Reviewers,
Thank you very much for giving us your feedback and advice on how to improve our article. We have revised the manuscript accordingly. Please find below our point-by-point replies to all of your comments and suggestions. All the modifications that we have made to the original manuscript and iconographies are detailed with our responses. Within the manuscript, the changes are highlighted in red to make them easier to view.
Point to point reply to Reviewer #1
The article analyzes the effect of MT on gait in PD by means of a systematic literature review and statistical analyses, comparing the gait outcomes of the intervention groups with the control groups. This work is relatively comprehensive and logical, and I think it can be published on Sensors.
Reply: Thank you.
Point to point reply to Reviewer #2
Everything looks fine now.
Reply: Thank you.
Point to point reply to Reviewer #3
The authors have revised some of the mistakes in the presentation of the study, which allows a better visualization of the analysis they have performed.
However, some doubts remain about the results, which given the characteristics of the studies do not seem entirely convincing, specifically in the link between the treatment implemented and the improvements observed.
This should perhaps be the subject of a reflexive and critical analysis on the part of the authors.
Reply: Thank you for your comments. We have added a reflexive and critical analysis as request.
Please see line 173 to 183: “Globally, there are remaining uncertainties regarding the findings of the included studies [49-51], as the nature of original studies raises questions regarding the causal pathways of the various applied treatments and the observed changes. Further studies involving the use of placebo control groups and assessment of patients’ expectation may lead to different interpretations regarding the effectiveness of manual therapy techniques [75].
Moreover, Terell et al. [51] showed that the use of osteopathic cranial manipulative medicine may improve gait kinematics in PD patients. However, as highlighted by Guillaud et al. [76] in a systematic review, caution is warranted when interpreting these results due to the absence of reliable diagnostic procedures and the lack of effectiveness. Quality assessment with Pedro Scale reinforced this caution because items concerning the participants or assessors blinding were not fulfilled in this article (see Table 2).”
Please also see the new references below:
[75] M. Molina-Álvarez, et al., « Manual Therapy Effect in Placebo-Controlled Trials: A Systematic Review and Meta-Analysis », Int J Environ Res Public Health, vol. 19, no 21, p. 14021, 2022, doi: 10.3390/ijerph192114021
[76] A. Guillaud, N. Darbois, R. Monvoisin, et N. Pinsault. « Reliability of Diagnosis and Clinical Efficacy of Cranial Osteopathy: A Systematic Review », PLoS One, vol. 11, no 12, p. e0167823, 2016, doi: 10.1371/journal.pone.0167823.
Reviewer 2 Report
Comments and Suggestions for Authors
Everything looks fine now.
Author Response

(The authors gave the same response as above.)

Reviewer 3 Report
Comments and Suggestions for Authors
The authors have revised some of the mistakes in the presentation of the study, which allows a better visualization of the analysis they have performed.
However, some doubts remain about the results, which given the characteristics of the studies do not seem entirely convincing, specifically in the link between the treatment implemented and the improvements observed.
This should perhaps be the subject of a reflexive and critical analysis on the part of the authors.
Author Response

(The authors gave the same response as above.)
